# The Reasonableness Behind Unreasonable Translation Capability of Large Language Model

**Tingchen Fu**$^{\heartsuit\diamondsuit*}$ **Lemao Liu**$^{\diamondsuit\dagger}$**Deng Cai**$^{\diamondsuit}$ **Guoping Huang**$^{\diamondsuit}$ **Shuming Shi**$^{\diamondsuit}$ **Rui Yan**$^{\heartsuit\dagger}$
$^{\heartsuit}$Gaoling School of Artificial Intelligence, Renmin University of China
$^{\diamondsuit}$Tencent AI Lab
`lucas.futingchen@gmail.com`  `redmondliu@tencent.com`  `ruiyan@ruc.edu.cn`

## Abstract

Multilingual large language models (LLM) trained on non-parallel data yield impressive translation capabilities. Existing studies demonstrate that incidental sentence-level bilingualism within pre-training data contributes to the translation abilities of large language models. However, it has been observed that the translation capabilities persist even when incidental sentence-level bilingualism is excluded from the training corpus. Therefore, in this study, we comprehensively investigate the question *why LLM can acquire translation capability without sentence-level bilingualism data.* To this end, we examine the impacts of word-level bilingualism data (i.e., word alignment data and code-switching data) or even purified monolingual data in addition to sentence-level bilingualism data. Through extensive experiments, we have made significant findings. It turns out that word alignment data plays a crucial role in enabling LLMs to acquire translation ability. Surprisingly, the translation signal derived from word alignment data is even comparable to that obtained from sentence-level bilingualism. Moreover, pre-training on purified monolingual data may enable a slight translation signal for LLM, thanks to the shared parameters in Transformer and some shared tokens across both source and target languages. Our code is available in `https://github.com/TingchenFu/ICLR24-TransContamination`.

## 1 Introduction

The performance of modern neural machine translation (NMT) systems is highly subject to the quality and volume of available parallel corpus. NMT systems trained in low-resource or unsupervised settings (Ravi & Knight, 2011; Artetxe et al., 2018) are usually less comparable to the ones trained in fully supervised setting (Lample et al., 2018; Lin et al., 2022; Garcia et al., 2021). However, recent multilingual large language models (LLMs), exemplified by ChatGPT (OpenAI, 2022) and GPT4 (OpenAI, 2023), seemingly defy conventional wisdom and exhibit unreasonable effectiveness in machine translation (García et al., 2023; Vilar et al., 2022; Zhang et al., 2023a). Intriguingly, these models are trained on non-parallel corpus, as opposed to the explicit parallel corpora commonly employed for NMT training. Therefore, it is imperative to comprehend *why LLMs enable translation by learning from non-parallel corpus.*

Concerning the reason why LLMs could learn to translate, a natural assumption is the existence of incidental sentence-level bilingualism in the pre-training corpus, which provides supervision signals and has been demonstrated sufficient to train an NMT system (Briakou et al., 2023). Nevertheless, Briakou et al. (2023) discover that PaLM (Chowdhery et al., 2022) still demonstrates non-trivial translation capability even after removing such incidental sentence-level bilingualism, indicating that it only partially accounts for LLM's translation capability. As a result, there must be *other sources contributing to the LLM's translation capability* beyond the presence of incidental sentence-level bilingualism.

---

$^{*}$ This work was done during an internship at Tencent AI Lab.
$^{\dagger}$Corresponding author: Lemao Liu (`redmondliu@tencent.com`) and Rui Yan (`ruiyan@ruc.edu.cn`).

Therefore in this study, we aim to comprehensively answer the in-depth question of why LLMs are capable of translation even without incidental sentence-level bilingualism. We hypothesize that the finer granularity of unintentional bilingualism, specifically word alignment data or code-switching data (exemplified in Table 1), may also play a substantial role. To this end, we propose to measure and compare the impact of unintentional bilingualism across various levels of granularity. Specifically, we first identify and collect three types of unintentional bilingualism data from existing multilingual corpora, namely sentence alignment, word alignment, and code-switching (§3). To quantify the effect of three types of bilingualism, it is too costly to train LLM from scratch and thereby we develop two computationally feasible methods as surrogates to measure the impact of data (§4). Subsequently, we apply the two surrogate methods to BLOOM-family model (Scao et al., 2022) with our collected three types of unintentional bilingual data and compare their effects on translation capacity (§5). Moreover, extensive experiments are conducted to glean insights into the impact of other factors (e.g., monolingual data, parameter-sharing, data volume) on the acquisition of translation capacity for LLM §5.

To summarize, our analysis yields two main findings:

- Aside from sentence alignment data, other forms of unintentional bilingualism also play an important role in assisting large language model to learn translation capability. In particular, we discover that word alignment data exhibit comparable or sometimes superior effectiveness in providing translation signals when compared with sentence alignment data.

- It is possible to observe a slight translation signal by pre-training solely on purified monolingual corpora in addition to unintentional bilingualism, thanks to some shared tokens (such as digits and symbols) and the shared parameters in model architecture across both source and target languages.

## 2 RELATED WORK

**Large Language Models for Translation.** Multilingual large language models, with Chat-GPT (OpenAI, 2022) and GPT4 (OpenAI, 2023) as representatives, have garnered heated attention for their impressive capacity in neural machine translation (Hendy et al., 2023; Zhu et al., 2023; Vilar et al., 2022; García et al., 2023) that is on par with not only commercial translation systems like Google Translate and Microsoft Translator (Hendy et al., 2023) but also the winner of WMT competition (Zhang et al., 2023a). Numerous previous works have devoted considerable efforts to evaluating and analyzing the translation abilities of these LLMs in both high and low resource languages (Hendy et al., 2023; Jiao et al., 2023b; Robinson et al., 2023), multilingual translation (Zhu et al., 2023), document-level translation (Wang et al., 2023; Karpinska & Iyyer, 2023). However, most existing works put emphasis on enhancing the performance of LLMs with fine-tuning (Xu et al., 2023a; Jiao et al., 2023a; Yang et al., 2023; Zeng et al., 2023; Zhang et al., 2023b) or seeking a better prompting strategy (Bawden & Yvon, 2023; He et al., 2023; Ghazvininejad et al., 2023; Lu et al., 2023), In contrast, little attention is paid on the question *why are these LLMs able to translate when they are only trained on multilingual corpus but witness no parallel data?*

**Data Contamination** Data contamination (Magar & Schwartz, 2022) refers to the possible overlap between the pre-training corpus and the benchmark for downstream evaluation (Brown et al., 2020). Data contamination can lead to inflated evaluation results and overestimation of the model's true performance since the model might inadvertently memorize the "leaked" information (Dodge et al., 2021; Lewis et al., 2021; Carlini et al., 2018) that was already present in the pre-training corpus. The phenomenon has been observed and reported on close-book QA (Lewis et al., 2021; Kandpal et al., 2022), numerical reasoning (Razeghi et al., 2022) and so on. On the other hand, Blevins & Zettlemoyer (2022) observe the mixing of other languages in "officially" English corpora (e.g., BookCorpus Zhu et al., 2015, and C4.en Raffel et al., 2020), christened "language contamination", may explain the acquisition of the language model's cross-lingual ability. Inspired by Blevins & Zettlemoyer (2022), we posit that language contamination, or in other words the mixing and coexisting of multiple languages, may also be present in non-parallel multilingual corpora and could potentially account for the translation capacity of multilingual LLMs. Most relevant to our work, Briakou et al. (2023) observe and extract unintentional bilingualism from the pre-training corpus of PaLM (Chowdhery et al., 2022) and examine the impact of various types of bilingualism. How-

ever, their experiments and analysis primarily involve sentence-level bilingualism on a closed-source LLM (Chowdhery et al., 2022), but overlook translation signals at a finer granularity.

## 3 Sentence Level and Word Level Bilingualism in Corpus

Usually derived from the Common Crawl[1] with a sophisticated data filtering pipeline, existing multilingual corpora such as mC4 (Xue et al., 2021), CC-100 (Conneau et al., 2020a), ROOTS (Laurençon et al., 2022b) and OSCAR (Ortiz Su'arez et al., 2019) are composed of multiple monolingual splits with their languages detected by language classification tools (Joulin et al., 2016). Nevertheless, due to the imperfections of the language classifier (Blevins & Zettlemoyer, 2022), a monolingual split may inadvertently contain some unintentional bilingual data. In this study, we focus on unintentional bilingualism between English and Chinese and identify three types of unintentional bilingualism present in three widely used multilingual corpora:

The first case is **sentence alignment (SA)**, wherein a sentence and its translation co-exist within close proximity in a document. Sentence alignment data constitutes a form of *sentence-level bilingualism* Analogously, **word alignment (WA)** pertains to the co-occurrence of one or more words (though not an entire sentence) and their translations within close proximity in a single document. Both sentence alignment and word alignment involve bilingual translation and differ solely in the granularity of translation. Additionally, we identify **code-switching (CS)**, which specifically refers to the co-occurrence of two languages within close proximity in a document, where the content in the two languages is semantically related rather than bearing a direct translation relationship. Both word alignment and code-switching are regarded as *word-level bilingualism*. Table 1 showcases three kinds of unintentional bilingualism.

| Type | Example |
|---|---|
| **Sentence Alignment** | *This news, like a light as an indescribable speed, In the blink of an eye it spread throughout the entire Martial Dragon Continent.*
*这个消息，如同光芒一般，以无法形容的速度，眨眼间就传遍了整个龙武大陆。*
This news was like a bullet, landed on the tranquil lake in the middle, instantly exploded! |
| **Word Alignment** | Beijing will procure RMB 80 million in social organization services.
*Beijing News (新京报), January 28, 2013* |
| **Code-Switching** | 上一篇(Previous Article)：New Polio Immunization Drive to Start in Nigeria's
下一篇(Next Article)：Hong Kong's Top Health Official Resigns Over SARS |

Table 1: Examples of three types of unintentional bilingualism found in mC4. For sentence alignment and word alignment, *the text in italics* is a parallel bilingualism in English and Chinese. For code-switching, the text highlighted in gray is our self-added translation for illustration.

To excavate three types of unintentional bilingualism, we develop a data mining pipeline. Briefly, to identify and gather unintentional bilingualism from a monolingual Chinese split: (1) We first search for alphabetic-composed fragments in each document with a regular expression. (2) The fragments recognized to be English by an off-the-shelf language detection tool (Joulin et al., 2016) are then translated into Chinese with an external commercial translation system (Huang et al., 2021). (3) Finally, we measure the similarity between the obtained translation and the nearby context of the found fragments and categorize the fragment together with its nearby context into three types of unintentional bilingualism accordingly. Aside from collecting unintentional bilingualism, for the purpose of experimental comparison, we also gather a "pure" monolingual Chinese (English) corpus by meticulously eliminating unintentional bilingualism as much as possible. More specifically, we again use regular expressions to remove any English letters (Chinese Characteristics) from the Chinese corpus (English corpus) in mC4. More details can be found in Appendix A.

The distribution of unintentional bilingualism in three corpora is depicted in Figure 1. As evidenced by the figure, unintentional bilingualism only accounts for a very small proportion (less than 5%) in the entire corpus. Nonetheless, given the vast scale of the corpus, the absolute quantity of contaminated documents is indeed non-trivial. Furthermore, as a common attribute spanning the three corpora, word alignment and code-switching predominantly comprise the primary components of unintentional bilingualism.

---

[1] https://commoncrawl.org/

| Dataset | Language | Test set | Example pool |
|---------|----------|----------|--------------|
| WMT21 | English-Chinese | newstest2021 (1948/1002) | newstest{2017,2018,2019} |
| FLORES-200 | English | eng_Latn.devtest (1012) | eng_Latn.dev (997) |
| | Chinese | zho_Hans.devtest (1012) | zho_Hans.dev (997) |
| | Catalan | cat_Latn.devtest (1012) | cat_Latn.dev (997) |
| | Eastern Panjabi | pan_Guru.devtest (1012) | pan_Guru.dev (997) |
| | Igbo | ibo_Latn.devtest (1012) | ibo_Latn.dev (997) |
| | Tswana | tsn_Latn.devtest (1012) | tsn_Latn.dev (997) |

Table 2: Statistics of our evaluation benchmarks. Numbers in brackets denote the number of instances.

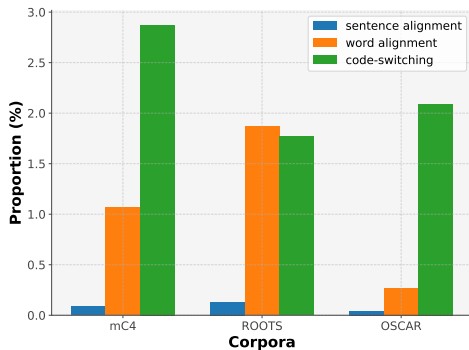

Figure 1: The proportion of documents containing three types of unintentional bilingual text in three corpora. The proportion is estimated by subset sampling.

| | | mC4.en | mC4.zh |
|---|---|--------|--------|
| sentence | # Doc | 210,931 | 2,462 |
| alignment | # Seq | 355,320 | 432 |
| word | # Doc | 658,643 | 1,972,764 |
| alignment | # Seq | 500,550 | 659,456 |
| code- | # Doc | 2,021,502 | 5,086,373 |
| switching | # Seq | 903,810 | 997,376 |

Table 3: The statistics of our mined unintentional bilingual text from mC4.en and mC4.zh. We concatenate all the unintentional bilingual data together with their nearby context in documents to form input sequences of fixed length. More details could be found in Appendix A.

## 4 METHODOLOGY TO QUANTIFY TRANSLATION CAPABILITY

To quantify and analyze the contribution of unintentional bilingual data, Briakou et al. (2023) primarily extract sentence alignment instances from the PaLM pre-training corpus, convert them into translation pairs $(x, y)$, thereby train an external sequence-to-sequence NMT model and draw comparison with another NMT model trained with official WMT parallel data to verify the effectiveness of sentence alignment data. Unfortunately, the strategy is not applicable when investigating the impact of other types of unintentional bilingualism since we can hardly convert word alignment data or code-switching data into translation pairs $(x, y)$ to train an external NMT system.

**Key Idea**   Alternatively, we do not train an external NMT model but directly train an LLM on different data sources and evaluate their performances as follows:

1. Train an LLM on our collected **X** data. X is one of three types of unintentional bilingualism, i.e., sentence alignment (SA), word alignment (WA), and code-switching (CS).
2. Train another LLM on a **X-rand** dataset as a comparison to X. X-rand data is randomly sampled from the original C4.en and C4.zh with the same number of examples as X. Note that it is contaminated with unintentional bilingualism and contains SA, WA and CS as shown in Figure 1.
3. Compare different LLMs (X vs. X-rand) with various metrics to measure their translation quality.

Accordingly, it can be inferred that: 1) if the LLM trained on X data performs better than that on X-rand data, then X contributes to LLM's translation capability; otherwise, X may have little influence on translation ability. 2) if the LLM trained on $X_1$ performs better than the models trained on $X_2$, then we can conclude that $X_1$ contributes more to LLM's translation capability than $X_2$ ($X_1$ or $X_2$ represents one of SA, WA or CS) and vice versa.

**Implementation**   However, a naive implementation of the above strategy is rather resource-intensive and infeasible in practice, as it involves the pre-training of multiple LLMs corresponding

to various data types. To put it into practice, we employ two surrogate methods that are more computationally efficient. The first approach, dubbed **post-training**, involves the continued training of the LLM under various data configurations based on their released checkpoints. The second approach, referred to as **pre-training**, is to pre-train a smaller LM with random initialization which serves as a simulation of training large language models from scratch. More implementation details can be found in Appendix C.

**Metrics**    Since $n$-gram metrics like BLEU (Papineni et al., 2002) are known to underestimate the performance of large language models (García et al., 2023), we mainly use COMET (Rei et al., 2020) and BLEURT (Sellam et al., 2020) to measure translation performance following Zhang et al. (2023a). Besides, we obtain a fine-grained and explainable evaluation of the translation performance employing InstructScore (Xu et al., 2023c;b), a compact yet competitive metric closely matching COMET in translation evaluation. Unfortunately, for the pre-trained smaller language models, their translation capability is not strong enough to yield readable translations through decoding, as evidenced in § 5.3 and it would be meaningless to measure the quality of these translations in terms of COMET and BLEURT. Instead, for evaluating these pre-trained models, we use a metric that is independent of decoding, i.e., perplexity for the conditional distribution of reference translation $y$ on the source $x$ to measure their translation ability.

## 5    WHY LLMs CAN TRANSLATE? AN EMPIRICAL ANALYSIS

### 5.1    EXPERIMENT SETUP

**Dataset**    If not otherwise specified, we use the unintentional bilingual data and purified monolingual data (i.e., excluding SA, WA and CS) from mC4.en and mC4.zh to perform the experiments, and their statistics are shown in Table 3. We use WMT21 news translation task (Akhbardeh et al., 2021) and the FLORES-200 (team et al., 2022) as our evaluation benchmarks, with their statistics presented in Table 2.

**Language Model**    In this study, we mainly focus on decoder-only multilingual LLM (Lin et al., 2022; Scao et al., 2022; Wei et al., 2023) that is not explicitly trained on parallel corpus and choose BLOOM (Scao et al., 2022) to perform our experiments for its diverse language category (45 natural languages and programme codes) and public availability.

**Prompt**    Given a test case $(x_{\text{test}}, y_{\text{test}})$, we randomly sample $n$ in-context translation examples $(x_1, y_1), (x_2, y_2), \ldots, (x_n, y_n)$ from the example pool (Table 2), and then insert into the following template borrowed from García et al. (2023) :

$$\{\texttt{source\_language}\} : \{x_1\}$$
$$\{\texttt{target\_language}\} : \{y_1\}$$
$$\cdots$$
$$\{\texttt{source\_language}\} : \{x_n\}$$
$$\{\texttt{target\_language}\} : \{y_n\}$$
$$\{\texttt{source\_language}\} : \{x_{\text{test}}\}$$
$$\{\texttt{target\_language}\} :$$

where `source_language` and `target_language` are the names of source and target languages.

### 5.2    POST-TRAINING EXPERIMENT ON BILINGUALISM

We first consider post-training BLOOM-7.1b and the experiment results on WMT21 are displayed in Table 4. As shown in the table, both sentence alignment data and word alignment data contribute to the translation ability, as they significantly outperform their X-rand counterparts in general. Notably, the overall performance of post-trained models is akin to the original BLOOM-7.1b model, which may be attributed to the marginal influence of post-training data in light of the pre-existing

| | ZH-EN | | | | EN-ZH | | | |
|---|---|---|---|---|---|---|---|---|
| | 3-shot | | 5-shot | | 3-shot | | 5-shot | |
| | COMET | BLEURT | COMET | BLEURT | COMET | BLEURT | COMET | BLEURT |
| BLOOM-7.1b | 59.58 | 37.21 | 60.38 | 38.01 | 79.84 | 57.87 | 80.34 | 58.58 |
| SA | 62.05* | 41.24* | 61.79* | 40.47* | 79.77 | 58.32* | 80.18 | 58.64 |
| SA-rand | 59.13 | 37.60 | 59.28 | 37.73 | 79.47 | 57.48 | 79.99 | 58.33 |
| WA | 58.36* | 36.34* | 58.15* | 35.75* | 79.59 | 57.61 | 80.11* | 58.46 |
| WA-rand | 56.21 | 32.91 | 56.51 | 33.32 | 79.48 | 57.42 | 79.86 | 58.14 |
| CS | 60.00* | 38.59* | 59.54* | 37.82* | 78.59 | 56.63 | 79.48 | 57.53 |
| CS-rand | 56.64 | 33.39 | 57.50 | 34.44 | 79.20 | 57.34 | 80.24 | 58.30 |

Table 4: Translation performance for post-training BLOOM-7.1b. CS = code-switching, WA = word alignment, SA = sentence alignment. Numbers marked with asterisk are significant improvements (t-test, $p < 0.05$) compared with the second-best model in the same block.

| | ZH-EN | | | | EN-ZH | | | |
|---|---|---|---|---|---|---|---|---|
| | 3-shot | | 5-shot | | 3-shot | | 5-shot | |
| | COMET | BLEURT | COMET | BLEURT | COMET | BLEURT | COMET | BLEURT |
| BLOOM-560m | 53.62 | 34.00 | 54.55 | 35.14 | 66.84 | 43.23 | 67.88 | 44.40 |
| SA | 61.55* | 43.14* | 61.57* | 43.04* | 69.27* | 46.32* | 69.98* | 47.14* |
| SA-rand | 54.87 | 36.41 | 55.26 | 36.60 | 61.80 | 38.58 | 63.71 | 40.33 |
| WA | 60.99* | 42.09* | 60.77* | 41.72* | 71.82* | 49.03* | 72.47* | 50.24* |
| WA-rand | 58.47 | 37.83 | 57.47 | 36.39 | 67.55 | 43.44 | 68.23 | 44.27 |
| CS | 59.02* | 39.66* | 59.22* | 39.90* | 68.24 | 45.41 | 69.35 | 46.60* |
| CS-rand | 58.43 | 38.56 | 57.95 | 37.49 | 68.53 | 44.70 | 69.26 | 45.27 |

Table 5: Translation performance for post-training smaller-scale LLM (BLOOM-560m). CS = code-switching, WA = word alignment, SA = sentence alignment. Numbers marked with asterisk are significant improvements (t-test, $p < 0.05$) compared with the second best model in the same block.

translation capabilities possessed by the original BLOOM-7.1b.[2] Therefore, for BLOOM-7.1b, the comparison should be limited to X and X-rand counterparts as the performance of post-training models is dominated by the original BLOOM-7.1b. In addition, 5-shot performance is usually better than 3-shot ones in general, which echoes prior findings (Zhang et al., 2023a) that more in-context examples usually help.

Therefore, we conduct similar post-training experiments on top of a smaller LLM (i.e., BLOOM-560m). Table 5 summarizes the experiment results. From this table, we can observe that both SA and WA significantly outperform their X-rand counterparts by a large margin in all cases, demonstrating that both SA and WA substantially contribute to LLM's translation ability. Notably, to our surprise, we find the effect of word alignment data is comparable or even superior to that of sentence alignment. We gauge one possible reason is that the number of WA examples contained in the pre-training corpus greatly exceeds the number of SA examples, as shown in Table 3. This unexpected discovery can further be used to elucidate an important phenomenon wherein the LLM's translation capability persists when sentence-level bilingualism is excluded from the training corpus — an observation previously noted in (Briakou et al., 2023) but lacking a clear explanation. In addition, we also find that CS outperforms CS-rand by a modest margin. This fact indicates that code-switching data imparts weak translation knowledge to LLMs.

## 5.3 PRE-TRAINING EXPERIMENT ON BILINGUALISM

Aside from the post-training experiment, we pre-train a self-implemented BLOOM-560m from scratch with our collected data as the simulation of training LLM. Owing to the constraints of our computational resources, we are regrettably unable to train a fully-converged BLOOM-560m[3] but alternatively train for a fixed number of updates, resulting in our self-implemented pre-trained mod-

---

[2]We present the results of the original BLOOM-7.1b only for reference but not for a basic baseline to compare since post-training may incur domain shift or hyper-parameter mismatching and thus lead to inferior performance than the original BLOOM-7.1b.

[3]Training a BLOOM-560m necessitates 92.61 days on 32 A100 GPUs.

| | ZH-EN | | | | EN-ZH | | | |
|---|---|---|---|---|---|---|---|---|
| | target | 1-shot | 3-shot | 5-shot | target | 1-shot | 3-shot | 5-shot |
| SA | 149.09 | 117.69 | 121.74 | 123.09 | 1303.59 | 523.11* | 525.32* | 527.93* |
| SA-rand | 85.50 | 110.85 | 109.13 | 109.00 | – | – | – | – |
| WA | 115.72 | 80.58* | 81.21* | 81.30* | 346.65 | 216.33* | 212.53* | 212.10* |
| WA-rand | 130.63 | 154.89 | 151.66 | 150.60 | 489.26 | 375.24 | 363.67 | 363.34 |
| CS | 138.36 | 129.82 | 131.35 | 132.37 | 343.39 | 270.67* | 268.45 | 273.18 |
| CS-rand | 91.34 | 112.20 | 109.21 | 108.21 | 351.53 | 281.94 | 269.61 | 266.18 |

Table 7: Translation performance for pre-training smaller-scale LLM (BLOOM-560m) in terms of perplexity. CS = code-switching, WA = word alignment, SA = sentence alignment. Numbers marked with an asterisk are significant improvements (t-test, $p < 0.05$) compared with the second-best model in the same block; The Numbers underlined are smaller translation perplexity than target language modeling perplexity. "–" denotes the number is above $2 \times 10^3$.

els being incapable of producing meaningful translations upon decoding. As is shown in Table 6, the majority of pre-trained models under various data settings exhibit only near-random[4] COMET scores that are much lower than the numbers in Table 5, indicating the potential risk of drawing a conclusion based solely on the comparison of the rather subpar translations upon decoding.

Alternatively, we use bilingual perplexity with various numbers of examples (1-shot/3-shot/5-shot) to measure their translation ability, which is obtained by concatenating the $y_{test}$ into the prompt outlined in §5.1 and computing the perplexity on $y_{test}$. Apart from that, we also present the monolingual perplexity of target language modeling (the "target" column in Table). In implementation, we solely input the $y_{test}$ into language models to compute its perplexity. From the information theory perspective (Xu et al., 2020), when bilingual perplexity is lower than monolingual perplexity, the LLM encapsulates positive mutual information between two languages and is able to employ information in source languages, which can be interpreted as translation capacity to a certain degree. Otherwise, it indicates that the language model treats the source language as noise and its translation capacity is too weak to be observed. [5]

| | ZH-EN | | EN-ZH | |
|---|---|---|---|---|
| | COMET | BLEURT | COMET | BLEURT |
| SA | 38.07 | 18.47 | 32.54 | 5.16 |
| SA-rand | 23.96 | 8.48 | 24.19 | 2.82 |
| WA | 35.96 | 16.36 | 41.22 | 3.73 |
| WA-rand | 30.35 | 6.25 | 33.29 | 2.45 |
| CS | 39.40 | 18.71 | 37.10 | 6.39 |
| CS-rand | 37.45 | 17.63 | 37.91 | 6.49 |
| random | 36.15 | 3.54 | 31.18 | 0.73 |

Table 6: Very weak translation performance of pre-training BLOOM-560m evaluated on WMT21 with 5 in-context examples. The near-random performance suggests the numbers might be noisy and thus unreliable.

We can observe that the effect of word alignment data is clearly evidenced by the reduced bilingual perplexity of WA in comparison to WA-rand. In the case of SA vs. SA-rand, SA-rand attains better multilingual perplexity in ZH-EN yet fails at EN-ZH direction, possibly because of the imbalanced proportion of Chinese data in sentence alignment (refer to Table 3). Notably, since the bilingual perplexity exceeds the monolingual ones, from the perspective of information theory we can conclude that SA-rand does not exhibit a translation capacity. In contrast, SA, WA, and CS all achieve positive mutual information, although CS does not outperform CS-rand in terms of multilingual perplexity. In summary, both WA and SA contribute significantly to LLM's translation capability, mirroring the findings of the post-training experiments.

## 5.4 TRANSLATION CAPABILITY FROM PURE MONOLINGUAL DATA

So far we have verified the role of unintentional bilingualism in enhancing the LLM's translation capabilities. Inspired from Pires et al. (2019); Artetxe et al. (2020), we pose a further question: Does the presence of unintentional bilingualism within the pre-training corpus constitute a prerequisite for translation ability? In other words, after eliminating unintentional bilingualism, can the LLM

---

[4]The "random" in Table 6 refers to directly outputting a random shuffle of source sentences in the test set as system hypothesis.

[5]Mathematically, if the perplexity of target language modeling $p(y)$ if higher than the conditional perplexity $p(y \mid x)$, then the mutual information $\mathcal{I}_p = H_p(y) - H_p(y \mid x)$ is positive (Xu et al., 2020).

| # step | ZH-EN | | | | EN-ZH | | | |
|---|---|---|---|---|---|---|---|---|
| | target | 1-shot | 3-shot | 5-shot | target | 1-shot | 3-shot | 5-shot |
| 4.5k | 145.54 | 155.12 | 148.94 | 147.00 | 562.75 | 804.49 | 727.77 | 711.32 |
| 7.5k | 114.55 | 141.42 | 132.69 | 129.77 | 450.33 | 517.05 | 477.31 | 511.61 |
| 10k | 66.13 | 90.49 | 83.75 | 81.87 | 242.68 | 200.21 | 182.51 | 179.54 |

Table 8: The translation capacity (in terms of perplexity) on WMT21 for pre-training smaller-scale LLM (BLOOM-560m) on purified monolingual data. The underlined numbers represent bilingual perplexity values that are lower than the monolingual perplexity in the target language.

| | target | 1-shot | 3-shot | 5-shot |
|---|---|---|---|---|
| Default | 242.68 | 200.21 | 182.51 | 179.54 |
| Sep Layer | 241.83 | 304.03 | 290.31 | 288.03 |
| Digit Substitution | 410.69 | 734.54 | 512.70 | 432.36 |

Table 9: The translation capacity (in terms of perplexity) on WMT21 EN-ZH for pre-training smaller-scale LLM (BLOOM-560m) on purified monolingual data.

acquire translation abilities simply by training on pure monolingual data in English and Chinese? To answer the question, we pre-train BLOOM-560m from scratch exclusively using pure monolingual data in English and Chinese. The experiment results are shown in Table 8.

We observe that the LLM exhibits positive mutual information on EN-ZH direction after training for 10k steps, proving that LLM could acquire a slight translation signal through pure monolingual data, though possibly conditioned on sufficient data and specific translation direction, which may be interesting to most researchers. To account for the counter-intuitive translation ability from pure monolingual corpora, we postulate two crucial factors: (1) **some common tokens (e.g., digits and symbols) shared across both languages**, which are mapped to shared space and can act as anchoring points to initially align their adjacent context and progressively extend to the entire semantic space (Pires et al., 2019; Conneau et al., 2020b). Meanwhile, (2) **the shared parameters in Transformer across two languages** may enable the model to detect the language-universal structures, thereby learning to align the representation of multiple languages with the help of anchor points, or unintentional bilingualism in our scenario (Artetxe et al., 2020; Dufter & Schütze, 2020).

To investigate the effect of shared tokens such as digits and symbols, we experiment with a variant **Digit Substitution** in which we substitute all digits in pure monolingual corpora with corresponding words in respective languages.[6] The results are shown in Table 9 and it becomes apparent that the positive mutual information on EN-ZH direction disappears, which primarily agrees with our hypothesis.

To verify the second factor regarding the impact of parameter-sharing architecture, we conduct an experiment comparing the performance of **Default** BLOOM-560m architecture with a variant in which the shared transformer layers are replaced with language-specific ones, dubbed **Sep Layer**. To be more specific, English data and Chinese data have respective transformer blocks and shared word embedding, as shown in Figure 2. The experiment results are shown in Table 9, substantiating the crucial role of parameter-sharing between languages, as the default variants exhibit significantly lower monolingual perplexity.

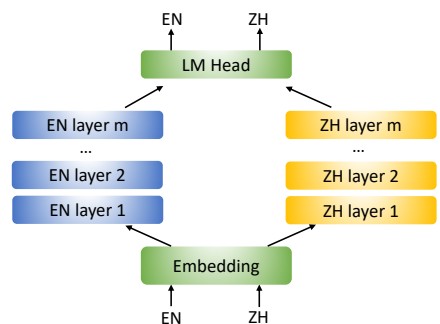

Figure 2: The model architecture of the Sep Layer variant. The modules in blue and yellow are exclusive for English and Chinese, respectively. The modules in green are shared modules of two languages.

---

[6]For example, we substitute "one" for "1" in English and "一" for "1" in Chinese.

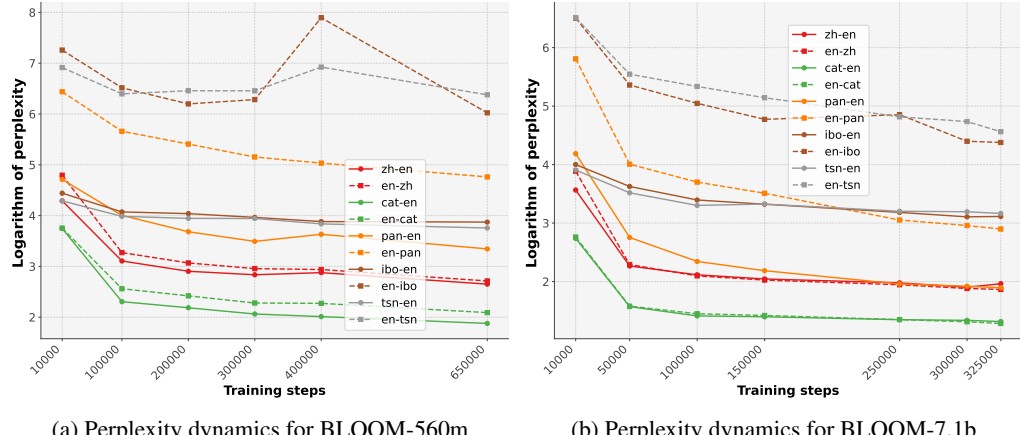

(a) Perplexity dynamics for BLOOM-560m      (b) Perplexity dynamics for BLOOM-7.1b

Figure 3: The perplexity dynamics of BLOOM model family on data size (training step) for both X-EN and EN-X.

## 5.5 EFFECTS OF TRANSLATION CAPABILITY ON DATA SIZE

To better understand how the translation ability of LLMs emerged, we examine the influence of pre-training data size utilizing the officially released intermediate checkpoints of the BLOOM-family model. [7] The trend for BLOOM-560m and BLOOM-7.1b on multiple language pairs (English ↔ Chinese/Catalan/Panjabi/Igbo/Tswana) are shown in Figure 3. Please refer to Appendix D for our rationale for choosing language pairs. From the figures, we can observe a similar trend:

1. The translation perplexity on a specific language is correlated to its makeup proportion in the pre-training corpus. The LLM achieves a lower perplexity for high-resource languages (e.g., Chinese, Catalan) compared to low-resource ones (e.g., Igbo, Tswana).
2. The translation ability experiences a surge at the early stage (approximately 1/6 of the whole training process) of training but plateaus or gradually increases thereafter. Note that this cannot be solely attributed to changes in learning rate since the warmup phase ends and the learning rate begins to decay at around 1/1000 of the whole training process;
3. Overall, X-EN and EN-X exhibit a similar trend for any language X. Except for one or two outliers, X-EN and EN-X usually surge, plateau, fluctuate, or decline in tandem, implying the two translation directions X-EN and EN-X may constitute a duality relationship (He et al., 2016).
4. The dynamics of BLOOM-560m and BLOOM-7.1b (and other variants shown in Appendix D) exhibit similar patterns and trends, despite the differing parameter scales. This implies that small-sized PLMs might share the same underlying mechanisms as LLMs when learning to translate.
5. Translation into low-resource languages is noticeably worse than from low-resource languages into English, showing how important target language modeling is in the estimation of translation quality. With more English data, models are able to produce better English output whereas low-resource language output is less polished.

## 6 CONCLUSION

Our study concentrates on the acquisition of translation capability in multilingual LLMs. To comprehend the origins of this capacity, we verify the presence of word alignment and code-switching data in three mainstream multilingual corpora. Our experiments emphasize that the finer granularity of unintentional bilingualism (i.e., word alignment data) yields a significant impact, which is comparable to or surpasses the effect of sentence alignment data. The translation ability of LLM still has room for improvement, particularly in low-resource languages (Robinson et al., 2023; Zhu et al., 2023) where LLM usually lags behind supervised translation model (team et al., 2022). As a possible remedy, our discoveries may inspire the research community in devising data augmentation or supervised fine-tuning techniques to address this challenge, which would also be our future work.

---

[7] https://huggingface.co/bigscience

REPRODUCIBILITY STATEMENT

All Our code, data and model checkpoints involved in this study will be released to facilitate related research. We present the details of our data collection and process pipeline in Appendix A. The implementation details are described in Appendix C with the selection of hyper-parameters in Table 16.

ETHICS STATEMENT

This paper will not pose any ethical problems. First, machine translation is an old task in natural language processing and representation learning with numerous papers about this task published at ICLR conferences. Second, all the corpora used in this paper have been used in previous papers. Our method should only be used to understand the translation capacity of multilingual large language models or other relevant research uses but not for any malicious purpose.

ACKNOWLEDGEMENT

This work was supported by National Natural Science Foundation of China (NSFC Grant No. 62122089), Beijing Outstanding Young Scientist Program NO. BJJWZYJH012019100020098, and Intelligent Social Governance Platform, Major Innovation & Planning Interdisciplinary Platform for the "Double-First Class" Initiative, Renmin University of China, the Fundamental Research Funds for the Central Universities, and the Research Funds of Renmin University of China.

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

# A MORE DATA COLLECTION DETAILS

Our experiments are conducted on the excavated data from the mC4 corpus (Xue et al., 2021). mC4 is a commonly used multilingual corpus, from which we randomly sample $192,715,980$ English documents and use all of $54,542,336$ Chinese documents to collect three types of unintentional bilingualism and "pure" monolingual English/Chinese corpora. Specifically,

1) To search for English fragments from Chinese documents, we first use the regular expression
`[A-Za-z0-9 _\,\.\;\:\'\"\?\!\-#\&]*[a-zA-Z]{3,}[A-Za-z0-9 _\,\.\;\:\'\"\?\!\-#\&]*`
to detect alphabetic-composed fragment. We use `fasttext` as our language detection tool. The fragment recognized to be English by `fasttext` with a prediction score over 90 is sent to TranSmart system[8] for translation.

2) Next, we use sacrebleu[9] (Post, 2018) to measure the similarity between the nearby context of the English fragment and the obtained translation. The fragment and its nearby context[10] are categorized as sentence alignment (SA) if the fragment contains at least 5 words and the sacreBLEU score is above 10%. Otherwise, it would be categorized as word alignment (WA) if the similarity is non-zero or code-switching (CS) if it is zero.

3) The procedure is almost the same when it comes to detecting unintentional Chinese text from English documents. The only difference is how we seek possible Chinese fragments. Specifically, we search for character-composed fragments whose Unicode lies between `\u4e00` and `\u9fff`, the interval corresponding to CJK unified ideographs.

4) Analogously, to obtain "pure" monolingual English/Chinese corpus for experimental use, we again use regular expressions to eliminate possible bilingual fragments from a "monolingual" corpus to obtain its purified version. Specifically, we get rid of all the strings matched by
`[A-Za-z0-9 _\,\.\;\:\'\"\?\!\-#\&]*[a-zA-Z]{3,}[A-Za-z0-9 _\,\.\;\:\'\"\?\!\-#\&]*`
from Chinese corpus and all strings whose Unicode lies within `\u2e80` and `\u9fff`[11] from an English corpus.

In our experiment, we first pre-process documents and group them into fixed-length sequences. In other words, sequences are assembled by concatenating and/or splitting documents to the appropriate length. Multiple short documents may compose only a single sequence, or a long document will be cut into a few sequences. Different from Briakou et al. (2023), we do not add a special document-boundary token.

## A.1 DATA QUALITY CONTROL

To check the quality of our collected unintentional bilingual data, we recruited three human annotators from our institute, each possessing a bachelor's degree and excelling in both languages. We randomly sample 100 instances from the SA/WA/CS data derived from the EN/ZH corpus and then inquire whether the presented texts align with the definitions of SA/WA/CS. The final decision is reached based on the majority vote of the three annotators. The annotators remain unaware of the origin of sentences so as to eliminate potential bias. The human evaluation results are shown in the Table 10.

|  | EN | ZH |
|---|---|---|
| sentence alignment | 87% | 90% |
| word alignment | 91% | 93% |
| code-switching | 90% | 92% |

Table 10: The human evaluation on the quality of our curated unintentional bilingual data.

---

[8] https://transmart.qq.com/

[9] https://github.com/mjpost/sacrebleu

[10] We define the nearby context or close proximity as the adjacent lines of the fragment.

[11] We expand the scope here including CJK strokes, CJK compatibility, CJK punctuation and so on to mitigate false negatives.

| | ZH-EN | | | | EN-ZH | | | |
|---|---|---|---|---|---|---|---|---|
| | 3-shot | | 5-shot | | 3-shot | | 5-shot | |
| | COMET | BLEURT | COMET | BLEURT | COMET | BLEURT | COMET | BLEURT |
| SA(threshold=15.0) | 60.76 | 42.33 | 60.64 | 42.21 | 69.08 | 46.51 | 69.69 | 47.15 |
| SA(threshold=15.0)-rand | 52.97 | 34.59 | 53.12 | 34.62 | 60.47 | 36.71 | 62.46 | 38.92 |
| SA(threshold=12.5) | 60.58 | 41.97 | 60.61 | 42.00 | 68.93 | 46.19 | 69.68 | 47.11 |
| SA(threshold=12.5)-rand | 50.55 | 37.73 | 50.59 | 34.01 | 59.89 | 37.19 | 62.27 | 39.40 |

Table 11: Translation performance for post-training smaller-scale LLM (BLOOM-560m) with different sentence alignment threshold. SA = sentence alignment.

| | zh-en | | | en-zh | | |
|---|---|---|---|---|---|---|
| | 1-shot | 3-shot | 5-shot | 1-shot | 3-shot | 5-shot |
| BLOOM-560m | 17.50 | 17.39 | 17.44 | 25.09 | 22.36 | 21.69 |
| SA | 15.02 | 14.87 | 14.87 | 23.37 | 22.31 | 22.26 |
| SA-rand | 17.38 | 17.36 | 17.36 | 37.22 | 31.73 | 30.83 |
| CS | 15.33 | 15.33 | 15.36 | 23.10 | 21.11 | 20.97 |
| CS-rand | 16.27 | 16.52 | 16.52 | 25.63 | 23.08 | 22.94 |
| WA | 11.37 | 11.50 | 11.62 | 18.51 | 17.63 | 17.60 |
| WA-rand | 16.52 | 16.66 | 16.65 | 25.64 | 22.81 | 22.64 |

Table 12: The perplexities on WMT21 after post-training BLOOM-560m with different data settings. CS = code switch, WA = word alignment, SA = sentence alignment.

## A.2 MORE ANALYSIS ON CLASSIFICATION THRESHOLD

To verify whether varying hyper-parameters for classifying sentence alignment yield different conclusions, we modify the similarity threshold for sentence alignment classification from 10.0 to 12.5 and 15.0 (measured in BLEU). Utilizing the updated sentence alignment thresholds, the experimental outcomes for post-training BLOOM-560m are displayed in Table 11.

As shown in the table, alterations to the threshold may impact specific translation performance on WMT21, but not our overall conclusion. In essence, our findings remain robust under varying similarity thresholds and are relatively unaffected by minor adjustments of the threshold.

## A.3 TRAIN-TEST OVERLAP

ROOTS corpus (Laurençon et al., 2022a) is only partially publicly available so it would be infeasible to accurately compute the overlap between the BLOOM pre-training data and the WMT21 test set. Therefore, we randomly sample 50 cases from WMT21 EN-ZH and ZH-EN respectively and approach the overlap proportion by manually searching similar passages in the pre-training corpus with the ROOTS fuzzy match tool (Piktus et al., 2023). Following previous work (Vilar et al., 2022; Chowdhery et al., 2022), we regard a test case as leaked if a sub-string of its target sentence with a minimum length of 15 tokens is contained in top-10 search results returned by the ROOTS fuzzy search tool. Through our analysis, the overlap proportion of both WMT21 EN-ZH and ZH-EN are less than 2.0%.

## B MORE EXPERIMENT RESULTS ON POST-TRAINING

We present the perplexity results for two post-training experiments (BLOOM-560m and BLOOM-7.1b) in Table 12 and Table 13, respectively. From the tables, we could see the perplexity results are consistent with those of BLEURT and COMET (Table 4 and Table 5), verifying that perplexity is an acceptable proxy for measuring translation ability.

Besides, we employ InstructScore (Xu et al., 2023c) to have a fine-grained and explainable evaluation of translation performance. The experiment results on post-training BLOOM-560m and BLOOM-7.1b are shown in Table 14 and Table 15 respectively.

| | ZH-EN | | | EN-ZH | | |
|---|---|---|---|---|---|---|
| | 1-shot | 3-shot | 5-shot | 1-shot | 3-shot | 5-shot |
| BLOOM-7.1b | 6.65 | 6.84 | 6.86 | 9.64 | 9.38 | 9.34 |
| SA | 6.31 | 6.62 | 6.76 | 9.75 | 9.64 | 9.62 |
| SA-rand | 6.40 | 6.65 | 7.22 | 10.44 | 9.85 | 9.95 |
| WA | 6.29 | 6.69 | 6.80 | 9.39 | 9.25 | 9.25 |
| WA-rand | 6.81 | 7.02 | 7.06 | 9.80 | 9.58 | 9.56 |
| CS | 6.60 | 6.79 | 6.88 | 9.85 | 9.60 | 9.58 |
| CS-rand | 6.69 | 6.92 | 6.95 | 9.96 | 9.85 | 9.76 |

Table 13: The perplexities on WMT21 after post-training BLOOM-7.1b under different data settings. CS = code switch, WA = word alignment, SA = sentence alignment.

| | ZH-EN | | EN-ZH | |
|---|---|---|---|---|
| | 3-shot | 5-shot | 3-shot | 5-shot |
| BLOOM-560m | -9.16 | -9.08 | -11.51 | -11.70 |
| SA | -9.01 | -9.09 | -11.95 | -11.91 |
| SA-rand | -9.32 | -9.32 | -11.98 | -12.12 |
| WA | -9.15 | -9.24 | -11.72 | -11.66 |
| WA-rand | -9.87 | -10.33 | -11.78 | -11.59 |
| CS | -9.25 | -9.05 | -11.74 | -11.63 |
| CS-rand | -9.43 | -9.74 | -11.75 | -11.75 |

Table 14: The InstructScore on WMT21 after post-training BLOOM-560m under different data settings. CS = code switch, WA = word alignment, SA = sentence alignment.

| | ZH-EN | | EN-ZH | |
|---|---|---|---|---|
| | 3-shot | 5-shot | 3-shot | 5-shot |
| BLOOM-7.1b | -9.88 | -9.89 | -10.51 | -10.45 |
| SA | -9.64 | -9.76 | -9.55 | -9.57 |
| SA-rand | -9.34 | -9.27 | -10.48 | -10.45 |
| WA | -9.76 | -9.76 | -9.71 | -9.61 |
| WA-rand | -10.30 | -10.40 | -10.73 | -10.64 |
| CS | -9.87 | -9.46 | -9.21 | -9.24 |
| CS-rand | -9.83 | -9.82 | -10.68 | -10.60 |

Table 15: The InstructScore on WMT21 after post-training BLOOM-7.1b under different data settings. CS = code switch, WA = word alignment, SA = sentence alignment.

| | post-training (BLOOM-560m) | pre-training (BLOOM-560m) | post-training (BLOOM-7.1b) |
|---|---|---|---|
| Precision | `float16` | `float16` | `float16` |
| Batch Size | 256 | 512 | 128 |
| Optimizer | AdamW | AdamW | AdamW |
| Adam ($\beta_1, \beta_2$) | (0.9,0.95) | (0.9,0.95) | (0.9, 0.95) |
| Learning Rate | 1e-5 | 3e-4 | 1e-4 |
| Sequence Length | 1024 | 1024 | 1024 |
| Warmup Step | 0 | 500 | 0 |
| Decay style | `cosine` | `cosine` | `cosine` |
| Min. Learning Rate | 0 | 0 | 0 |
| Weight Decay | 1e-1 | 1e-1 | 1e-1 |
| Gradient clip | 1.0 | 1.0 | 1.0 |
| LoRA rank | NA | NA | 8 |
| LoRA $\alpha$ | NA | NA | 16 |

Table 16: The hyper-parameters for post-training and pre-training.

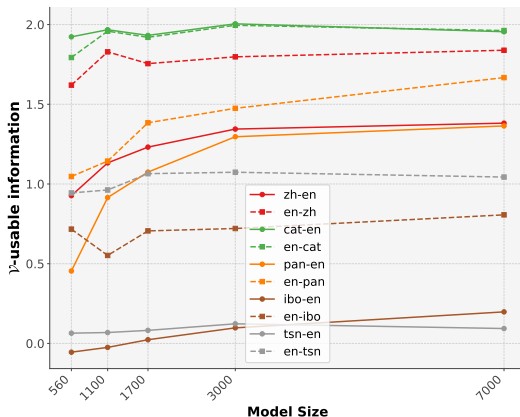

Figure 4: The perplexity dynamics of BLOOM model family on model size for both X-EN and EN-X.

## C   MORE IMPLEMENTATION DETAILS

Our experiments are conducted on a cloud Linux server with Ubuntu 16.04 operating system. The codes are written in Python 3.10 using the code from huggingface library[12]. The GPU type is Nvidia Tesla V100 with 32GB GPU memory.

The detailed hyper-parameter settings for post-training and from-scratch training are shown in Table 16. Note that for post-training of BLOOM-7.1b, without loss of generality, we use the LoRA (Hu et al., 2022) as a parameter-efficient training technique rather than full-parameter training. We apply the low-rank adaptation for the query, key, value and output projection matrices in the self-attention module within every transformer layer. We train the model for one epoch for both post-training and pre-training. To ensure a fair comparison, we maintain the size and language composition of X-rand to be consistent with X. Specially, when preparing X-rand data, the sampling ratio between C4.en and C4.zh is in alignment with the composition proportion shown in Table 3.

For prompting, we randomly sample in-context examples from the candidate pool. For decoding, we use greedy search with a minimal generation length of 5. It is possible that with a more sophisticated prompting and decoding algorithm we may get better results but the decoding algorithm or prompting strategy is not the focus of this study.

---

[12]`https://huggingface.co/`

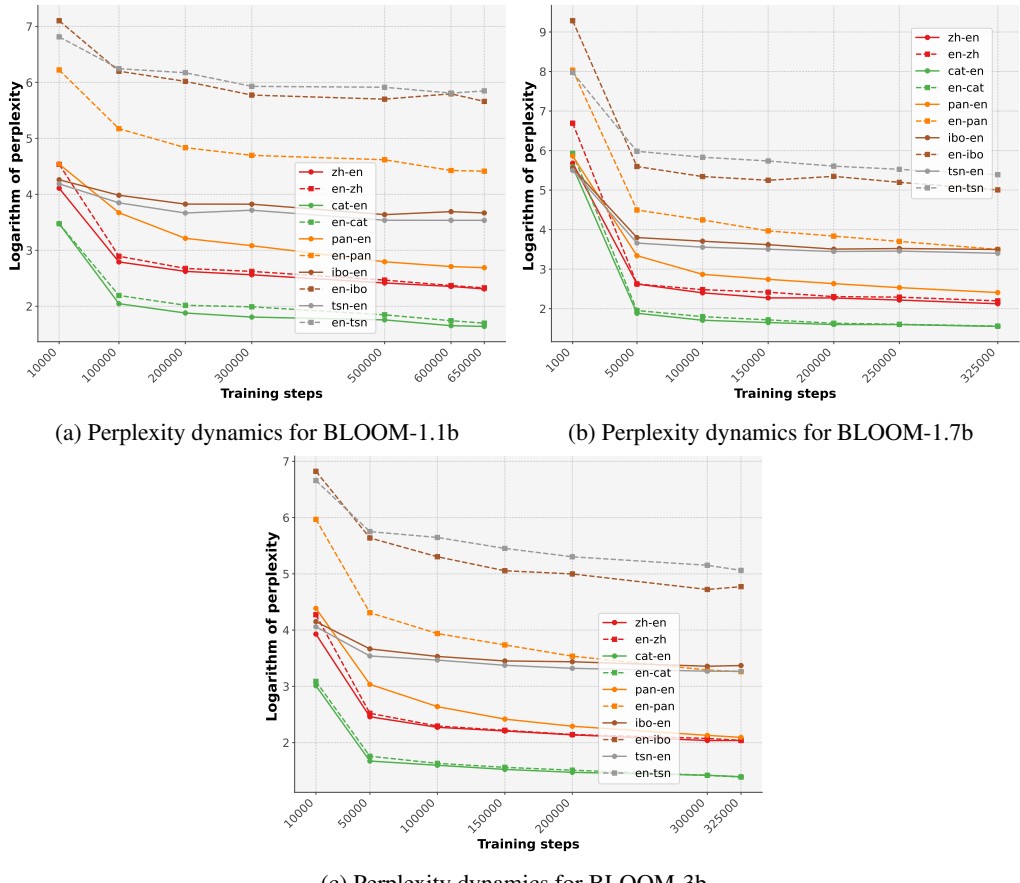

(a) Perplexity dynamics for BLOOM-1.1b        (b) Perplexity dynamics for BLOOM-1.7b

(c) Perplexity dynamics for BLOOM-3b

Figure 5: The perplexity dynamics of BLOOM model family on data size (training step) for both X-EN and EN-X.

# D   MORE EXPERIMENTS RESULTS ON LEARNING DYNAMICS

As BLOOM models are trained on English (33%) and other 45 natural languages, we rank these languages according to their make-up proportion in ROOTS (Laurençon et al., 2022b) and thereby select 5 languages, namely Simplified Chinese (1st, 18%), Catalan (10th, 1.2%), Eastern Punjabi (19th, 0.1%), Igbo (28th, 0.001%) and Setswana (37th, 0.0001%) to perform to (X-EN) and from (EN-X) English translation experiments on the FLORES-200 (team et al., 2022) benchmark. The effect of data size (training steps) for BLOOM-560m and BLOOM-7.1b are shown in Figure 3a and Figure 3b respectively. Besides, we also track the learning dynamics of BLOOM-1.1b, BLOOM-1.7b and BLOOM-3b, The results are shown in Figure 5. In addition, we plot the trend of translation capacity (measured in multilingual perplexity) with the model size on Figure 4.

We can observe that the shape of the curve initially exhibits an obvious decline from 560m to 1.7b, followed by a period of gradual tapering off. Notably, for low-resource languages (e.g., Igbo and Setswana) which constitute a small portion of the training corpus, solely expanding the model size to enhance their translation quality is a suboptimal approach. This strategy necessitates a considerably larger model size but might only yield marginal returns, which echoes previous findings (Kaplan et al., 2020; Kandpal et al., 2022) to a certain degree.

