# OpenReview forum: "The Reasonableness Behind Unreasonable Translation Capability of Large Language Model"
_ICLR.cc/2024/Conference — ICLR 2024 poster_

### Official Review · Reviewer_6ach · 2023-10-30

**Soundness:** 3 good
**Presentation:** 3 good
**Contribution:** 3 good
**Rating:** 6
**Confidence:** 4

**Summary:**

This paper explored the reasons for the emergent translation capabilities of LLM from the perspective of data composition and provided empirical evidence to support its findings.

**Strengths:**

The paper innovatively delved into the emergent translation abilities of LLM by analyzing its training data composition.

**Weaknesses:**

The evaluation methods have its limitations.

There are flaws in the methodology, suggesting that the prompts might influence the results.

At its core, it's an ablation study at the data level.

**Questions:**

1. How can we be certain that it's related to numbers and punctuation? If we replace numbers with their unique expressions in different languages, such as 'one' for '1', 'two' for '2'; '壹' for '1', '贰' for '2' (Traditional Chinese); '一' for '1', '二' for '2' (Simplified Chinese), ensuring that each language's corpus is entirely unrelated, would the translation capability still exist?

2. Is the spontaneous alignment capability of large models based on the power of the attention framework? Regardless, does joint training of an LLM with two or more languages always generalize to translation capabilities, given a fixed number of languages in the data? This point is crucial for data augmentation methods in LLM.

3. Compared to supervised NMT models, the translation capability of large models is more evident in their flexible interactions, producing translations that better meet requirements. This flexibility also leads to the unpredictability of the translation outputs from large models. Even a minor change in a prompt can result in different translations. In light of this paper, how do the authors view this issue?

4. If possible, please consider using the 'Instruct Score' from EMNLP 2023 (Instructscore: Towards Explainable Text Generation Evaluation with Automatic Feedback) as a metric.

5. The title is quite bold, but I don't believe the content of the paper fully justifies it. For instance, if the focus is on the data composition of LLM, deeper issues like data leakage (where training data includes test data or text similar to test data) should be considered. The 'bloomz' dataset, which is open-source and includes translation tasks, doesn't seem to have been addressed in the paper. LLMs are trained on vast amounts of data, so there's a need to check for (partial) overlap with the test set.

6. Has there been any consideration that an excessive amount of bilingual data might dilute the pre-trained knowledge in LLM and impair its translation capability?

---

> ### Author Response · Authors · 2023-11-22
> **Thanks for your review! (part 1/3)**
>
> We are grateful for your time and effort in reviewing our work. The constructive and insightful suggestions would definitely help us improve the work. We would like to address your weaknesses and questions one by one as below and we are looking forward to your feedback or further comments. Your support is of great importance to us.
>
>
> **Q1: How can we be certain that it's related to numbers and punctuation?**
>
> **ANSWER**:
> Thanks for your suggestion! Following your advice, we substitute all digits in purified bilingual data with words in respective languages (e.g., "one" for $1$ in English and so on) and repeat our experiments. The experiment results are shown in the Table below and updated in our submission (Section 5.4):
>
>
> |         |  ZH-EN |        |        |        |  EN-ZH |        |        |        |
> |---------|:------:|:------:|:------:|:------:|:------:|:------:|:------:|:------:|
> |         | target | 1-shot | 3-shot | 5-shot | target | 1-shot | 3-shot | 5-shot |
> | 4.5k    | 174.42 | 187.75 | 185.38 | 188.20 | 662.53 | 871.74 | 797.75 | 744.06 |
> | 7.5k    | 114.34 | 126.31 | 130.49 | 122.63 | 414.13 | 736.75 | 528.85 | 456.46 |
> | 10k     | 105.69 | 121.84 | 119.52 | 117.46 | 410.69 | 734.54 | 512.70 | 432.36 |
>
> From the table, we can see that the positive mutual information on EN-ZH direction disappears once we remove all the digits in the pre-training data. Besides, we notice that the monolingual/bilingual perplexity is worse than that trained on the original purified data, possibly because replacing all the digits with English words or Chinese characters would cause a change in the pre-training data distribution. The experiment results primarily agree with our hypothesis that the existence of co-occurring digits might be instrumental in LLM obtaining translation ability from purified bilingual data.  However, to draw a more definitive conclusion, more experiments are needed and we would rather leave it as a topic for future exploration.
>
> **Q2: At its core, it's an ablation study at the data level. Is the spontaneous alignment capability of large models based on the power of the attention framework?**
>
> **ANSWER**: Honestly, our work is indeed a sort of ablation study from the data-driven perspective, following previous work[1]. Our work provides an explanation of the underlying mechanism by identifying the existence of unintentional bilingualism at finer granularity (which is omitted by [1]) and quantitatively analyzing their effect in language model pre-training. It is risky to draw conclusions on the power of attention framework based on our experiments since the question is not the focus of our work. But would like to work on how the design of model architecture including attention mechanism affects LLM's translation capacity in the future.
>
>
> **Q3: Regardless, does joint training of an LLM with two or more languages always generalize to translation capabilities, given a fixed number of languages in the data?**
>
> **ANSWER**: Frankly speaking, we cannot assure that LLM with two or more languages will always generalize to translation ability, which requires enumerating all language pairs and thus is infeasible. Alternatively, we perform experiments on English and Chinese as two representatives since they are the top $2$ languages in the language makeup of BLOOM pre-training data and they almost share no scripts. We believe that our findings are agnostic to the specific languages and could be extrapolated to other language pairs.

---

> ### Author Response · Authors · 2023-11-22
> **Thanks for your review! (part 2/3)**
>
> **Q4: There are flaws in the methodology, suggesting that the prompts might influence the results. Even a minor change in a prompt can result in different translations. In light of this paper, how do the authors view this issue?**
>
> **ANSWER**: Thanks for your reminder! Our choice of prompting follows previous work[2][3] and we totally agree that different wording in prompt indeed causes differences in translation performance.  To investigate the variance caused by different prompting, we adopt the most competitive prompt reported in [4]. Specifically, we format a given test case and $n$ randomly samped in-context examples $(x_1,y_1), (x_2,y_2), (x_3,y_3) ... (x_n,y_n)$ into the following template.
>
> {source_language}: {$x_1$} = {target_language}: {$y_1$}
>
> {source_language}: {$x_2$} = {target_language}: {$y_2$}
>
> {source_language}: {$x_3$} = {target_language}: {$y_3$}
>
> ...
>
> {source_language}: {$x_n$} = {target_language}: {$y_n$}
>
> {source_language}: {$x_{test}$} = {target_language}:
>
> The experiment results on post-training BLOOM-560m are shown in the table below:
>
> |            |  ZH-EN |        |        |        |  EN-ZH |        |        |        |
> |------------|:------:|:------:|:------:|:------:|:------:|:------:|:------:|:------:|
> |            | 3-shot |        | 5-shot |        | 3-shot |        | 5-shot |        |
> |            | COMET  | BLEURT | COMET  | BLEURT | COMET  | BLEURT | COMET  | BLEURT |
> | BLOOM-560m | 54.07  | 31.42  | 54.71  | 32.29  | 66.18  | 39.25  | 67.26  | 41.02  |
> | SA         | 60.10  | 40.77  | 60.23  | 41.35  | 68.59  | 44.82  | 69.32  | 45.88  |
> | SA-rand    | 54.53  | 35.01  | 54.30  | 35.40  | 62.23  | 37.07  | 63.07  | 38.21  |
> | WA         | 60.87  | 42.07  | 60.67  | 41.78  | 72.07  | 49.22  | 72.67  | 49.9   |
> | WA-rand    | 54.49  | 33.94  | 55.55  | 33.78  | 66.61  | 40.27  | 66.3   | 40.93  |
> | CS         | 58.15  | 38.65  | 58.33  | 39.10  | 67.87  | 43.88  | 69.4   | 45.46  |
> | CS-rand    | 55.90  | 34.91  | 56.86  | 35.98  | 65.51  | 39.25  | 67.32  | 41.45  |
>
> The experiment on post-training BLOOM-7.1b is shown in the table below:
> |            |  ZH-EN |        |        |        |  EN-ZH |        |        |        |
> |------------|:------:|:------:|:------:|:------:|:------:|:------:|:------:|:------:|
> |            | 3-shot |        | 5-shot |        | 3-shot |        | 5-shot |        |
> |            | COMET  | BLEURT | COMET  | BLEURT | COMET  | BLEURT | COMET  | BLEURT |
> | BLOOM-7.1b | 61.69  | 38.37  | 61.63  | 38.43  | 80.03  | 58.21  | 80.41  | 58.71  |
> | SA         | 64.78  | 42.92  | 64.39  | 42.28  | 79.85  | 58.44  | 79.89  | 58.05  |
> | SA-rand    | 61.52  | 39.53  | 59.96  | 37.25  | 79.57  | 57.65  | 79.76  | 58.09  |
> | WA         | 58.24  | 33.95  | 57.25  | 32.40  | 79.88  | 58.01  | 79.92  | 58.37  |
> | WA-rand    | 54.74  | 29.19  | 56.54  | 30.79  | 79.62  | 57.78  | 79.76  | 57.69  |
> | CS         | 61.90  | 33.70  | 61.69  | 39.80  | 79.07  | 57.25  | 79.32  | 57.79  |
> | CS-rand    | 56.64  | 33.39  | 57.50  | 34.44  | 79.20  | 57.34  | 80.24  | 58.30  |
>
> As is shown in the tables, turning to a different prompt indeed brings fluctuations in translation performance, but our conclusion remains. In other words, our finding is relatively insensitive to different prompting templates.
>
> **Q5: The evaluation methods have their limitations. If possible, please consider using the Instruct Score from EMNLP 2023**
>
> **ANSWER**: Instruct Score[5] is a brilliant idea providing a solution to explainable and learnable text generation evaluation. Following your advice, we use the instruction score to measure the translation ability, and the results on post-training BLOOM-560m are shown below:
>
> |            |  ZH-EN |        |  EN-ZH |        |
> |------------|:------:|:------:|:------:|:------:|
> |            | 3-shot | 5-shot | 3-shot | 5-shot |
> | BLOOM-560m | -9.16  | -9.08  | -11.51 | -11.70 |
> | SA         | -9.01  | -9.09  | -11.95 | -11.91 |
> | SA-rand    | -9.32  | -9.32  | -11.98 | -12.12 |
> | WA         | -9.15  | -9.24  | -11.72 | -11.66 |
> | WA-rand    | -9.87  | -10.33 | -11.78 | -11.59 |
> | CS         | -9.25  | -9.05  | -11.74 | -11.63 |
> | CS-rand    | -9.43  | -9.74  | -11.75 | -11.75 |
>
>
> The evaluation results on BLOOM-7.1b are shown in the table below:
>
> |            |  ZH-EN |        |  EN-ZH |        |
> |------------|:------:|:------:|:------:|:------:|
> |            | 3-shot | 5-shot | 3-shot | 5-shot |
> | BLOOM-7.1b | -9.88  | -9.89  | -10.51 | -10.45 |
> | SA         | -9.64  | -9.76  | -9.55  | -9.57  |
> | SA-rand    | -9.34  | -9.27  | -10.48 | -10.45 |
> | WA         | -9.76  | -9.76  | -9.71  | -9.61  |
> | WA-rand    | -10.30 | -10.40 | -10.73 | -10.64 |
> | CS         | -9.87  | -9.46  | -9.21  | -9.24  |
> | CS-rand    | -9.83  | -9.82  | -10.68 | -10.60 |
>
> We have incorporated the evaluation results (Appendix B) and a brief introduction of Instruct Score[5] (Section 4)  in our updated version of submission with the revision part highlighted in blue.

---

> ### Author Response · Authors · 2023-11-22
> **Thanks for your review! (part 3/3)**
>
> **Q6: The 'bloomz' dataset, which is open-source and includes translation tasks, doesn't seem to have been addressed in the paper. LLMs are trained on vast amounts of data, so there's a need to check for (partial) overlap with the test set.**
>
> **ANSWER**: Thanks for your advice. We need to clarify that our experiments do not use the BLOOMZ model or its instruction tuning dataset (P3 or xP3). Instead, we use the BLOOM model trained on ROOTS corpus[6] which does not explicitly contain parallel corpus. ROOTS corpus is only partially publicly available so it would be infeasible to accurately compute the overlap between the BLOOM pre-training data and the WMT test set. Therefore, we randomly sample $50$ cases from WMT21 EN-ZH and ZH-EN respectively and estimate the overlap proportion by manually searching similar passages in the pre-training corpus with the ROOTS fuzzy match tool[7]. Following previous work[3], we regard a test case as leaked if a substring of its target sentence with a minimum length of $15$ tokens is contained in the top-10 search results returned by the ROOTS fuzzy search tool. Through our analysis, the overlap proportion of both WMT21 EN-ZH and ZH-EN is less than 2.0%.
>
> We have incorporated the examination of train-test overlap in our updated version of submission (Appendix A.3) with the revision highlighted in blue.
>
> **Q7: Has there been any consideration that an excessive amount of bilingual data might dilute the pre-trained knowledge in LLM and impair its translation capability?**
>
> **ANSWER**: In theory, the unintentional bilingual data is not supposed to dilute the translation knowledge of LLM (but may impair the knowledge for other downstream tasks) and we can observe a promotion in translation performance after post-training on unintentional bilingual data for most experiment results in Table 4 and Table 5. Moreover, compared with the 1.6T pre-training data of BLOOM, our gathered unintentional bilingual data is rather minuscule in volume.
>
>
> [1]Briakou, Eleftheria, Colin Cherry, and George Foster. "Searching for Needles in a Haystack: On the Role of Incidental Bilingualism in PaLM's Translation Capability." arXiv preprint arXiv:2305.10266 (2023).
>
> [2]Vilar, David, et al. "Prompting palm for translation: Assessing strategies and performance." arXiv preprint arXiv:2211.09102 (2022).
>
> [3]Garcia, Xavier, et al. "The unreasonable effectiveness of few-shot learning for machine translation." International Conference on Machine Learning. PMLR, 2023.
>
> [4]Bawden, Rachel, and François Yvon. "Investigating the translation performance of a large multilingual language model: the case of bloom." arXiv preprint arXiv:2303.01911 (2023).
>
> [5]Xu, Wenda, et al. "Instructscore: Towards explainable text generation evaluation with automatic feedback." arXiv preprint arXiv:2305.14282 (2023).
>
> [6]Laurençon, Hugo, et al. "The bigscience roots corpus: A 1.6 tb composite multilingual dataset." Advances in Neural Information Processing Systems 35 (2022): 31809-31826.
>
> [7]Piktus, Aleksandra, et al. "The roots search tool: Data transparency for llms." arXiv preprint arXiv:2302.14035 (2023).

---

> > ### Comment · Reviewer_6ach · 2023-11-22
> >
> > The authors have responded well to the questions posed. Although there remains some confusion regarding the data-based ablation experiments, the perspective offered in the paper is commendable and encouraging. I have adjusted my rating to a 6.

---

> > > ### Author Response · Authors · 2023-11-22
> > > **Thanks for your feedback!**
> > >
> > > We are more than delighted to receive the positive feedback and the acknowledgment of the perspectives provided in the paper. Again, we are grateful for your efforts and expertise. Your constructive comments and your recognition of our response to the questions are highly appreciated!

---

### Official Review · Reviewer_stf2 · 2023-10-31

**Soundness:** 3 good
**Presentation:** 4 excellent
**Contribution:** 3 good
**Rating:** 6
**Confidence:** 4

**Summary:**

This submission addresses an important and relevant question in recent machine translation research: how do large language models manage to perform so well on multiple language pairs without being trained specifically on bilingual (parallel) data? It outlines three different types of unintentional bilingual data (UBD): sentence and word alignment, as well as code switching, and analyse their influence on machine translation performance, suggesting that word-level alignment may play a larger role than was previously thought, possibly because it is much more prevalent than sentence alignment.

**Strengths:**

The underlying question has been addressed before, suggesting that the presence of incidental sentence alignment in "supposedly" monolingual data plays a large role in LLM performance on MT. This submission pushes the investigation by looking at word-level alignment and code-switched data which as far as I know is novel.

The positive impact of word alignment is demonstrated in a number of experiments showing improvements in either translation quality or perplexity. The hypothesis that the larger amount of word alignment data (as compared to sentence-aligned) allows it to be as useful as sentence-aligned data is well supported by the experiments.

MT quality is evaluated using neural metrics rather than BLEU, which aligns with recent results from the WMT22 metrics task (https://aclanthology.org/2022.wmt-1.2/).

Some results are presented with significance tests of the differences, yay!

**Weaknesses:**

The UBD (sentence & word aligned + code switched) is extracted using an automated methods with arbitrary parameters that seem fairly ad hoc (eg 10 BLEU poins threshold, appdx A). This raises the question of how good that data is... Are sentence-aligned segments even aligned sentences? At 10 BLEU point, this is not very clear. This data is at the core of the argument of the paper, better quality control would make a more convincing case.

Experiments use mostly smaller models, as well as surrogate methods, and some models are not even converged. Clearly the amount of computation is significant. However, would one make conclusion on a chemical process from a reaction that has not completed? These imperfect or incomplete experimental conditions make the conclusions less convincing. The switch to perplexity does not help -- the claims in Section 5.3 are only mildly convincing if it is not possible to extract minimally useful translations from the models.

Section 5.4 would benefit from a comparison with a model using UBD data somehow -- this would allow to gauge whether the performance in Table 8 is getting remotely close to acceptable translation quality.

**Questions:**

p.8 -- Shouldn't perplexity de lower in Table 9 (using UBD) than in Table 10 (purified) -- Can you comment on why it would be the opposite?

Section 5.2: Although WA is better than WA-rand in Table 4, it is also inferior to both SA and CS, often by a large margin. Could you better support the claim that "the effect of word alignment data is comparable or even superior to that of sentence alignment"?

An additional final point in Section 5.6. could be that translation *into* low resource languages is noticeably worse than *from* low resource language into English, showing how important fluency is in the estimation of translation quality. With more EN data, models are able to produce better English output whereas low resource language output is less polished.

Misc/typos:
"... translation quality is less comparable to NMT systems trained with parallel data." is a bit ambiguous as low-resource systems are typically trained on parallel data as well, but much less of it.

"other forms of unintentional bilingualism also plays" -> play

"Sine" (p.5) -> Since

"CS outperforms its X-rand counterparts" could more clearly be "CS outperforms CS-rand"

---

> ### Author Response · Authors · 2023-11-22
> **Thanks for your review! (part 1/2)**
>
> We are very excited to receive your comment approving the novelty of our findings and the solidity of our experiment design. We are grateful for your time and effort in reviewing our work. We carefully read through your insightful and helpful comments and we would like to address weaknesses as well as questions one by one:
>
> **W1: This data is at the core of the argument of the paper, better quality control would make a more convincing case.**
>
> **ANSWER**: Thanks for your suggestion. We totally agree that data quality control is very important for our setup. Therefore, we perform a human evaluation of the quality of our collected corpus. Specifically, we gather judgments from three in-house annotators by randomly sampling $100$ cases from SA/WA/CS data collected from EN/ZH corpora respectively and asking them whether the shown texts agree with the definition of SA/WA/CS. The qualification of each case is determined by the major vote of three annotators. The annotators are volunteers from our institute, each possessing a bachelor's degree and excelling at both languages. They are unknown of the source of these sentences. The experiment results are shown in the table and updated in our submission (Appendix A.1):
>
> |                    | EN  | ZH  |
> |--------------------|-----|-----|
> | sentence alignment | 87% | 90% |
> | word alignment     | 91% | 93% |
> | code-switching     | 90% | 92% |
>
>
> Besides, to verify whether varying hyperparameters for classifying sentence alignment yield different conclusions, we modify the similarity threshold for sentence alignment classification from $10.0$ to $12.5$ and $15.0$ (measured in BLEU). Utilizing the updated sentence alignment thresholds, the experimental outcomes for post-training BLOOM-560m are displayed in the tables that follow:
>
>
> |                         |  ZH-EN |        |        |        |  EN-ZH |        |        |        |
> |-------------------------|:------:|:------:|:------:|:------:|:------:|:------:|:------:|:------:|
> |                         | 3-shot |        | 5-shot |        | 3-shot |        | 5-shot |        |
> |                         | COMET  | BLEURT | COMET  | BLEURT | COMET  | BLEURT | COMET  | BLEURT |
> | SA(threshold=12.5)      | 60.58  | 41.97  | 60.61  | 42.00  | 68.93  | 46.19  | 69.68  | 47.11  |
> | SA(threshold=12.5)-rand | 50.55  | 37.73  | 50.59  | 34.01  | 59.89  | 37.19  | 62.27  | 39.40  |
>
>
>
> |                         |  ZH-EN |        |        |        |  EN-ZH |        |        |        |
> |-------------------------|:------:|:------:|:------:|:------:|:------:|:------:|:------:|:------:|
> |                         | 3-shot |        | 5-shot |        | 3-shot |        | 5-shot |        |
> |                         | COMET  | BLEURT | COMET  | BLEURT | COMET  | BLEURT | COMET  | BLEURT |
> | SA(threshold=15.0)      | 60.76  | 42.33  | 60.64  | 42.21  | 69.08  | 46.51  | 69.69  | 47.15  |
> | SA(threshold=15.0)-rand | 52.97  | 34.59  | 53.12  | 34.62  | 60.47  | 36.71  | 62.46  | 38.92  |
>
> As shown in the tables, alterations to the threshold may impact specific translation performance, but not our overall conclusion. In essence, our findings remain robust under varying similarity thresholds and are relatively unaffected by minor adjustments of the threshold.
> Your advice definitely helps to improve our work and we have incorporated your concern and our answer in our updated submission (Appendix A.2).
>
>
> **W2: Experiments use mostly smaller models, as well as surrogate methods, and some models are not even converged.**
>
> **ANSWER:** Thanks for your advice. Limited computational resources are indeed a concern when designing our experiment. However, from another standpoint, BLOOM-560m and BLOOM-7b1 are the two most widely used versions within the BLOOM series, as evidenced by download statistics from the Huggingface Hub. Consequently, we have selected BLOOM-560m and BLOOM-7b1 as representatives. Training a fully converged BLOOM-560m model necessitates 92.61 days on 32 A100 GPUs, according to BLOOM's technical report [1], which surpasses the computation budget of many institutions. As a result, our surrogate method is more replicable and we promise to release our code to facilitate relevant research.

---

> ### Author Response · Authors · 2023-11-22
> **Thanks for your review! (part 2/2)**
>
> **W3: Section 5.4 would benefit from a comparison with a model using UBD data somehow.**
>
> **ANSWER**: Thanks for your advice! We pre-train a BLOOM-560m model on a blend of unintentional bilingual data
> to draw a comparison with the one trained on purified data. The results are shown as below:
>
>
> |             |  ZH-EN |        |        |        |  EN-ZH |        |        |        |
> |-------------|:------:|:------:|:------:|:------:|:------:|:------:|:------:|:------:|
> |             | target | 1-shot | 3-shot | 5-shot | target | 1-shot | 3-shot | 5-shot |
> | 4.5k        | 98.95  | 38.69  | 38.82  | 39.21  | 300.50 | 85.85  | 86.24  | 89.21  |
> | 7.5k        | 89.35  | 27.98  | 28.26  | 28.27  | 264.35 | 60.11  | 59.09  | 59.72  |
> | 10k         | 65.49  | 18.42  | 18.70  | 18.69  | 178.28 | 34.50  | 34.32  | 34.60  |
>
> From the table, the effect of unintentional bilingual data is evident. When comparing the results with Table 8 in section 5.4, it becomes evident that translation capacity acquired from the purified (uncontaminated) bilingual corpora is very weak though it does exist.
>
>
>
> **Q1: Shouldn't perplexity be lower in Table 9 (using UBD) than in Table 10 (purified)?**
>
> **ANSWER**: We apologize for any ambiguity. However, Table 8 and Table 9 should not be directly compared, as they differ in hyperparameters and language composition. In the experiment presented in Table 9, the data is randomly sampled from the union of mC4.en and mC4.zh, where English data prevails. Conversely, in the experiment depicted in Table 10, equal amounts of English and Chinese data are employed.
>
>
> **Q2: Although WA is better than WA-rand in Table 4, it is also inferior to both SA and CS, often by a large margin. Could you better support the claim that "the effect of word alignment data is comparable or even superior to that of sentence alignment"?**
>
> **ANSWER**: Our primary conclusions are derived from Table 4, Table 5, and Table 7. We observe that, in the EN-ZH direction for BLOOM-7.1b (Table 4), several values are lower than those for SA and CS. However, in Table 5, particularly in the EN-ZH direction, WA is comparable to or even exceeds SA. Consequently, we deduce that WA also plays a significant role in the acquisition of translation ability.
>
>
> **Q3**: Additional point in Section 5.6 and other misc/typos.
>
> **ANSWER**: Thanks for the advice! we have incorporated your advice into our updated version of submission with the revision part highlighted in blue.
>
>
> [1]Workshop, BigScience, et al. "Bloom: A 176b-parameter open-access multilingual language model." arXiv preprint arXiv:2211.05100 (2022).

---

### Official Review · Reviewer_5xLr · 2023-11-01

**Soundness:** 3 good
**Presentation:** 3 good
**Contribution:** 3 good
**Rating:** 5
**Confidence:** 3

**Summary:**

Multilingual large language model has shown impressive translation capabilities, while the model is trained on non-parallel data. This paper is motivated by a question of "Why can multilingual large language model can learn to translate without parallel data?" The authors report that there exist unintentional bilingualism in the training data that can be categorized into 3 types of bilingualism: 1) sentence alignment, 2) word alignment, and 3) code-switching and experimentally showed that, with these bilingualism, a large language model can learn to translate. Specifically, word alignment data provides the model with translation signals when compared with sentence alignment data. These findings will be helpful in data collection, data augmentation to improve translation capability of a large language model, since the translation ability lags behind a general translation model that is trained on parallel data.

**Strengths:**

- Extensive experiments are conducted, to address the main question of "Why can multilingual large language model can learn to translate without parallel data?"
- The authors provide detailed empirical analyses to identify the underlying mechanism of translation capability in multilingual large language model. They also discuss the cases where single monolingual data are available with different sizes. This part might be helpful on how to improve low-resource language translation quality in the language model
- The paper is well organized and clearly described in most parts.

**Weaknesses:**

- The authors focus on English and Chinese data to identify unintentional bilingualism types, and I was wondering if any other bilingualism types exist when you check the other languages.
- This paper looks interesting in light of the empirical analyses of bilingualism's role in translation capabilities; however, the question still remains on how to effectively enhance translation capability further against the supervised translation models.

**Questions:**

- In Table 3, why is #seq smaller than #Doc? The paper describes "We concatenate all the sentences in documents to form input sequences of fixed length" and I assume that #seq would be larger in #Doc.
- The authors focus on English and Chinese data to identify unintentional bilingualism types, and I was wondering if any other bilingualism types exist when you check the other languages.

- In Tables 9 and 10, please consider rephrasing "the translation performance" since both tables report PPL, not general translation performance metrics such as BLEU or COMET.
- n-gram -> $n$-gram

---

> ### Author Response · Authors · 2023-11-22
> **Thanks for your review!**
>
> Thanks for your time and expertise invested in reviewing our work! we are encouraged to receive the recognition of extensive experiment and analysis. We carefully read through your constructive and valuable suggestions and we would like to answer your questions one by one as below. We hope we can address your weaknesses as well as questions and we would be grateful for your support, which means a lot to us.
>
>
> **W1：The authors focus on English and Chinese data to identify unintentional bilingualism types, and I was wondering if any other bilingualism types exist when you check the other languages.**
>
> **ANSWER**: Thanks for your suggestion. Our work is inspired by [1], which identifies $4$ types of language contamination (unintentional bilingualism) phenomenon in more than $50$ languages. We use their subdivision of language contamination as a reference and identify three types of unintentional bilingualism in our work.
>
> Following your advice, we perform data mining on mC4.fr searching for unintentional bilingualism between English and French. We randomly sampled 160,000 French documents and 270,000 English documents and the proportion of documents with three types of unintentional bilingualism are shown in the Table below.
>
> |                    | mC4.en | mC4.fr |
> |--------------------|--------|--------|
> | sentence alignment | 0.1   | 0.4    |
> | word alignment     | 43.8  | 88.0     |
> | code-switching     | 84.2   | 84.5   |
>
> We can observe that a great portion of documents contain unintentional bilingualism (between English and French). Note that our detection and mining of unintentional bilingualism is subject to the accuracy of the language detection tool.  We primarily chose English and Chinese data as two representatives because of their high proportion in the pre-training corpus of BLOOM. Meanwhile, it is technically easy to detect unintentional bilingualism since they almost share no common scripts.
>
> **W2: However, the question still remains on how to effectively enhance translation capability further against the supervised translation models.**
>
> **ANSWER**: Thanks for your concern. Enhancing the translation capacity is crucial to the downstream application of large language model. However, our work's primary objective and contribution lie in tracing the origin and mechanisms of LLM's translation ability from a data-driven standpoint. As a by-product, we observe that post-training on unintentional bilingualism typically results in improved translation performance. Motivated by this phenomenon, one potential approach could involve incorporating more bilingual or cross-lingual data into the pre-training corpus or the supervised fine-tuning corpus, as explored by [2]. Consequently, we believe our findings will inspire researchers within the community to further enhance translation ability.
>
>
> **Q1: In Table 3, why is #seq smaller than #Doc? The paper describes "We concatenate all the sentences in documents to form input sequences of fixed length" and I assume that #seq would be larger in #Doc.**
>
> **ANSWER**: Sorry for the ambiguity. We actually concatenate all the unintentional bilingual data (SA/WA/CS) together with their nearby context in documents to form an input sequence of fixed length. So it is possible that the unintentional bilingual sentences from multiple documents constitute a single input sequence of fixed length ($1024$ in our experiment). We have corrected the caption in the updated version (highlighted in blue) to avoid misunderstanding.
>
>
> **Q2: Rephrasing and correcting typos**
>
> **ANSWER**: Thanks for your advice. We have incorporated your advice into our updated version of submission with the revision highlighted in blue.
>
>
> [1]Blevins, Terra, and Luke Zettlemoyer. "Language contamination helps explain the cross-lingual capabilities of English pretrained models." arXiv preprint arXiv:2204.08110 (2022).
>
> [2]Zhu, Wenhao, et al. "Extrapolating Large Language Models to Non-English by Aligning Languages." arXiv preprint arXiv:2308.04948 (2023).

---

### Official Review · Reviewer_ZA8F · 2023-11-01

**Soundness:** 3 good
**Presentation:** 3 good
**Contribution:** 3 good
**Rating:** 6
**Confidence:** 3

**Summary:**

The paper studies the translation ability in multilingual LLMs, showing that the presence of sentence alignment, word alignment and code-switching data may contribute maybe an important role in explaining this amazing ability of multilingual LLMs. It is interesting that word alignment data yields a significant impact that surpasses the effect of sentence alignment data in certain cases (e.g. for post-training smaller-scale LLM). Despite several cons I see from the work, I think this work is a nice and very timing contribution to the community and thus recommend for an acceptance.

**Strengths:**

* The findings from the paper are interesting and may shed light on the amazing translation ability from LLM. I enjoy reading the work a lot.

**Weaknesses:**

My biggest concern about the work is that while it is true that maybe the presence of sentence alignment, word alignment and code-switching data contribute an important role in explaining this amazing ability of multilingual LLMs. We just don't know how important they are and is there any other reason (e.g. the presence of other stuff) that is actually even way more important than the three. I don't think the paper provide any data points on this.

Another smaller concern is that I see some paragraphs are just half-baked from curiosity perspective. For instance, section 5.4 shows that after eliminating unintentional bilingual data, the translation ability is still there, apparently. But there is no data points on why that happens, just some postulating about the reasons and that is it.

The final weakness point of the paper to me is the presentation in Section 5.5. I could not follow exactly what "the shared transformer layers in the BLOOM-560m model". To make the work self-contained I think the paper should at least present some high level details about the shared transfomer layers before presenting how this may influence the effect of unintentional bilingualism.

**Questions:**

- I get lost at random baseline in Table 6. Please elaborate more.
- what is the precision/recall of "WA", "SA", "CS" classifier?

---

> ### Author Response · Authors · 2023-11-22
> **Thanks for your review! (part 1/2)**
>
> We are grateful for your acknowledgment that our work sheds light on the amazing translation ability of large language models. We are encouraged by your recognition. We read through your insightful comments, which definitely help us improve the work. We would like to address your weaknesses as well as questions one by one as follows:
>
> **W1: We just don't know how important they are and is there any other reason (e.g. the presence of other stuff) that is actually even way more important than the three?**
>
> **ANSWER**: Good question! We recognize that there may be additional factors within pre-training data, beyond the three types of inadvertent bilingualism, that contribute to translation capabilities. The experiments in Section 5.4 confirm the presence of such factors in the purified data. However, compared to unintentional bilingualism, these other factors have a relatively minor impact. As illustrated in the table below, we post-train BLOOM-560m using either a blend of the three types of unintentional bilingual data or the "residual data" sampled from the complementary portion of unintentional bilingualism within the entire pre-training corpus to evaluate their respective effects. (Note that residual data does not equal purified data, since the former may contain unrecognized unintentional bilingualism.) To ensure a fair comparison, the "residual data" utilized in our experiments is consistent with the unintentional bilingual data in terms of size and language makeup.
>
> |                              |  ZH-EN |        |        |        |  EN-ZH |        |        |        |
> |------------------------------|:------:|:------:|:------:|:------:|:------:|:------:|:------:|:------:|
> |                              | 3-shot |        | 5-shot |        | 3-shot |        | 5-shot |        |
> |                              | COMET  | BLEURT | COMET  | BLEURT | COMET  | BLEURT | COMET  | BLEURT |
> | unintentional bilingual data | 59.74  | 39.94  | 60.25  | 41.12  | 70.78  | 47.40  | 71.59  | 48.03  |
> | residual data                | 56.91  | 36.43  | 57.48  | 37.02  | 64.45  | 37.02  | 65.16  | 38.90  |
>
> From the table, it becomes evident that the influence of other factors pales in comparison to that of unintentional bilingualism. Hence, we can deduce that unintentional bilingualism is the most predominant factor, surpassing the others in effect.
>
> ---
> **W2: There are no data points on why training on purified training data would lead to translation ability, just some postulating about the reasons and that is it.**
>
>
> **ANSWER**: Thanks for your advice.
> Training on purified training data would lead to relatively weak translation ability. This is verified by our experiments: the translation ability is only partially observed in the EN-ZH direction, but it is not clear in the ZH-EN direction, as shown in Table 8. As a result, pinpointing the origin of such a weak translation capacity is very challenging because of its weak signal. Furthermore, the experiment referenced in our response to W1 also indicates that the LLM's translation capacity primarily stems from unintentional bilingualism, while the remaining data is of less importance. Therefore, focusing on unintentional bilingualism instead of purified data would be more valuable to understanding the translation ability of large language models. However, to render our analysis more comprehensive and self-contained, we substitute all digits in purified bilingual data with English words or Chinese characters  (e.g., "one" for $1$, "two" for $2$ in English and "一" for $1$ in Chinese) and repeat our experiments, following the advice of reviewer 6ach. The experiment results are shown in the Table below and are updated in our submission (Table 9):
>
>
> |         |  ZH-EN |        |        |        |  EN-ZH |        |        |        |
> |---------|:------:|:------:|:------:|:------:|:------:|:------:|:------:|:------:|
> |         | target | 1-shot | 3-shot | 5-shot | target | 1-shot | 3-shot | 5-shot |
> | 4.5k    | 174.42 | 187.75 | 185.38 | 188.20 | 662.53 | 871.74 | 797.75 | 744.06 |
> | 7.5k    | 114.34 | 126.31 | 130.49 | 122.63 | 414.13 | 736.75 | 528.85 | 456.46 |
> | 10k     | 105.69 | 121.84 | 119.52 | 117.46 | 410.69 | 734.54 | 512.70 | 432.36 |
>
> From the table, we can see that the positive mutual information on EN-ZH direction disappears once we remove all the digits in the pre-training data. Besides, we notice that the monolingual/bilingual perplexity is worse than that trained on the original purified data, possibly because replacing all the digits with English words or Chinese characters would cause a change in the pre-training data distribution. The experiment results primarily agree with our hypothesis that the existence of co-occurring digits might be instrumental in LLM obtaining translation ability from purified bilingual data. However, to draw a more definitive conclusion, more experiments are needed and we would rather leave it as a topic for future exploration.

---

> > ### Author Response · Authors · 2023-11-22
> > **Thanks for your review! (part 2/2)**
> >
> > **W3: I could not follow exactly what "the shared transformer layers in the BLOOM-560m model". To make the work self-contained I think the paper should at least present some high-level details about the shared transformer layers**
> >
> > **ANSWER**: We apologize for the oversimplification. Compared with the original 24-layer BLOOM model, the Sep-Layer version has 12 layers for English data and 12 layers for Chinese data in parallel, thus providing distinct computational pathways. In other words, no transformer layers will be trained on English corpus and Chinese corpus simultaneously. To better illustrate the model architecture, we have added a figure in our updated version (Figure 3).
> >
> >
> > **Q1: I get lost at random baseline in Table 6. Please elaborate more.**
> >
> > **ANSWER**: We apologize for any ambiguity. To construct the "random" baseline, we directly utilize the shuffled source sentences from the WMT21 test set as system output.
> > For instance, let's assume the test set consists of three pairs ${(x_1,y_1), (x_2,y_2), (x_3,y_3)}$. We take a random permutation of the source sentences, such as $[x_2,x_3,x_1]$, as the system output, even though the gold standard reference should be $[y_1,y_2,y_3]$. The aim of this experiment is to expose the unreliability of decoding-based evaluation metrics, namely COMET and BLEURT when the LLM's translation ability is suboptimal.
> > As demonstrated in Table 6, almost all model variants exhibit comparable BLEURT and COMET scores to the "random" baseline, highlighting the potential risk of relying solely on these two metrics for drawing conclusions.
> >
> >
> > **Q2:what is the precision/recall of "WA", "SA", "CS" classifier?**
> >
> > **ANSWER**: Good question! We collect judgments from three bilingual annotators within our institution, each possessing a bachelor's degree, by randomly sampling 100 instances from the SA/WA/CS data derived from the EN/ZH corpus. We then inquire whether the presented texts align with the definitions of SA/WA/CS. The ultimate decision is reached based on the majority vote of the three annotators, who remain unaware of the origin of sentences. The results are shown in the Table below:
> >
> > |                    | EN  | ZH  |
> > |--------------------|-----|-----|
> > | sentence alignment | 87% | 90% |
> > | word alignment     | 91% | 93% |
> > | code-switching     | 90% | 92% |
> >
> > The data quality control is updated in our submission (Appendix A.1) and your advice is definitely of great significance to our work.
> >
> > [1]Briakou, Eleftheria, Colin Cherry, and George Foster. "Searching for Needles in a Haystack: On the Role of Incidental Bilingualism in PaLM's Translation Capability." arXiv preprint arXiv:2305.10266 (2023).
> >
> > [2]Blevins, Terra, and Luke Zettlemoyer. "Language contamination helps explain the cross-lingual capabilities of English pre-trained models." arXiv preprint arXiv:2204.08110 (2022).

---

### Meta-Review · Area_Chair_SQZj · 2023-12-15

**Metareview:**

This paper addresses an important and relevant question: why are LLMs capable of doing translation without being trained specifically on bilingual (parallel) data. It outlines three different types of unintentional bilingual data (UBD): sentence and word alignment, and code switching. The authors analyze their influence on MT, suggesting that word-level alignment may play a larger role than was previously thought, possibly because it is much more prevalent than sentence alignment. The underlying question has been addressed before, suggesting that the presence of incidental sentence alignment in "supposedly" monolingual data plays a large role in LLM performance on MT. This submission pushes the investigation by looking at word-level alignment and code-switched data in more details (it was also part of the original paper [1]). The findings from the paper are interesting and may shed light on the amazing translation ability from LLMs. Extensive experiments are conducted. The experimental setup is sound and results are well presented. During the discussion period, the authors addressed all concerns from the reviewers and updated the paper accordingly. E.g. They performed a human evaluation to investigate the precision of the data mining procedure and added the results to the paper.

[1] https://arxiv.org/abs/2305.10266

**Justification For Why Not Higher Score:**

This is an incremental work and its strengths are more experimental evidence on why LLM can translate.

**Justification For Why Not Lower Score:**

There are no major reasons why this should not get accepted.

---

### Decision · Program_Chairs · 2024-01-16

Accept (poster)